# Effect of Boric Acid on the Ionization Equilibrium of α-Hydroxy Carboxylic Acids and the Study of Its Applications

**DOI:** 10.3390/molecules28124723

**Published:** 2023-06-12

**Authors:** Rongxiu Qin, Haiyan Chen, Rusi Wen, Guiqing Li, Zhonglei Meng

**Affiliations:** 1Guangxi Key Laboratory of Superior Timber Trees Resource Cultivation, Guangxi Forestry Research Institute, Nanning 530002, China; qrx20151112@126.com (R.Q.); chujin2005@163.com (H.C.); luse-rose@163.com (R.W.); liguiqing2107413@163.com (G.L.); 2Institute of Chemical Industry of Forest Products, Chinese Academy of Forestry (CAF), Nanjing 210042, China

**Keywords:** boric acid, α-hydroxycarboxylic acid, ionization equilibrium, fatty acid esterification, green synthesis

## Abstract

To investigate the synergistic catalytic effects of boric acid and α-hydroxycarboxylic acids (HCAs), we analyzed and measured the effects of the complexation reactions between boric acid and HCAs on the ionization equilibrium of the HCAs. Eight HCAs, glycolic acid, D-(−)-lactic acid, (R)-(−)-mandelic acid, D-gluconic acid, L-(−)-malic acid, L-(+)-tartaric acid, D-(−)-tartaric acid, and citric acid, were selected to measure the pH changes in aqueous HCA solutions after adding boric acid. The results showed that the pH values of the aqueous HCA solutions gradually decreased with an increase in the boric acid molar ratio, and the acidity coefficients when boric acid formed double-ligand complexes with HCAs were smaller than those of the single-ligand complexes. The more hydroxyl groups the HCA contained, the more types of complexes could be formed, and the greater the rate of change in the pH. The total rates of change in the pH of the HCA solutions were in the following order: citric acid > L-(−)-tartaric acid = D-(−)-tartaric acid > D-gluconic acid > (R)-(−)-mandelic acid > L-(−)-malic acid > D-(−)-lactic acid > glycolic acid. The composite catalyst of boric acid and tartaric acid had a high catalytic activity—the yield of methyl palmitate was 98%. After the reaction, the catalyst and methanol could be separated by standing stratification.

## 1. Introduction

α-Hydroxy carboxylic acids (HCAs), such as citric acid and malic acid, play an important role in life activities. The mentioned acids are involved in tricarboxylic acid cycle reactions. Boron is widely distributed on the earth and is an indispensable trace element for animals and plants [1,2]. Due to the electrical absorption of boron atoms, boric acid can form complex complexes with hydroxy-containing compounds as well as 1:1 and 1:2 complexes with HCAs [3]. The structure, reaction equilibrium, reaction kinetics, and thermodynamics of the complexes formed by boric acid and HCA have been extensively investigated by researchers [4,5,6,7]. Activation of acid by boric acid has been successfully applied in synthetic organic chemistry [8]. L-(+)-tartaric acid is a chiral polyhydroxyl compound which can form L-(+)-tartrate–boric acid with a ring structure through a complexation acid with boric acid in methanol solutions when the ratio of L-(+)-tartrate to boric acid is 2:1. This ratio increases the difference between the chiral complexation acid and the two enantiomers in terms of spatial matching and improves the chiral recognition [9]. In addition, the electrical conductivity and rotation of the complex formed by tartaric acid and boric acid are increased [10,11]. The hydration reaction of α-pinene catalyzed by HCA and boric acid can increase the conversion rate of α-pinene [12]. In the synthesis of isobornyl acetate and isoborneol catalyzed by HCA and boric acid composite catalysts from camphene, HCA and boric acid show significant synergistic catalytic effects [13]. Because the composite catalyst composed of HCA and boric acid can be used for both aqueous and non-aqueous reactions, it is beneficial to use it for the methylation reactions of fatty acids.

Fatty acid methyl ester (FAME) has been widely used in energy fields, pesticides, medicine, the chemical industry, and other fields [14]. The traditional preparation of FAME is often performed in the presence of acid catalysts such as H_2_SO_4_, HCl, H_3_PO_4_, or organic sulfonic acids [15]. However, the inherent corrosion and toxicity of these catalysts limit their use in continuous production [16,17]. The use of solid acid catalysts such as metal oxides [18], zeolites [19], and heteropolyacids [20] makes waste management of the system, product separation, and recycling of the catalyst more convenient, but solid acid catalysts suffer from low specific surface areas and low catalytic activities [21]. Supercritical catalysis is a new method for the synthesis of fatty acid methyl ester, which has advantages such as high reaction efficiencies and simple process flows, but it has strict requirements on the equipment [22]. Lipase is used as an efficient biocatalyst to catalyze the synthesis of fatty acid methyl esters. The reaction conditions are mild, the product is easy to isolate and purify, and the process is green and produces low levels of pollution. However, lipase is more sensitive to methanol and is easily poisoned, which leads to deactivation [23]. The research on catalysts with low costs, high catalytic activities, and easy recovery is of great significance for fatty acid esterification reactions.

This paper investigates the synergistic catalytic mechanism of boric acid and HCA by analyzing and measuring the effects of the complexation reactions between boric acid and HCAs on the ionization equilibrium of the HCAs. The selected catalyst was used for the esterification reaction of palmitic acid to synthesize methyl palmitate.

## 2. Results and Discussion

### 2.1. Influence of Boric Acid on pH of Hydroxy Carboxylic Acid (HCA) Aqueous Solution

Boric acid has an effect on the ability of HCAs to ionize protons through the reaction of boric acid with HCA hydroxyl groups to form various complexes, thus promoting a positive shift in the ionization equilibrium. The determination of the pH for different molar ratios of boric acid to HCA provides insight into the effects of the complexation reactions on the ionization equilibria and informs the design of catalysts. We used HCAs, including glycolic acid (GA), citric acid (CA), L-(+)-tartaric acid (L-(+)-TA), D-(−)-tartaric acid (D-(−)-TA), D-(−)-lactic acid (LA), (R)-(−)-mandelic acid (MA), L-(−)-malic acid (H_2_Mi), and D-gluconic acid (GlcA).

#### 2.1.1. Measurement of pH after Addition of Boric Acid to Aqueous HCA Solution

The pKa1 of boric acid (9.24) is much larger than those of HCAs (3.04–3.86). If the influence of the self-ionization of boric acid on the pH is ignored, the influence of boric acid on the ionization equilibrium of different HCAs can be investigated by measuring the pH after adding boric acid for a certain HCA concentration, as shown in Figure 1, Figure 2 and Figure 3.

#### 2.1.2. Rates of pH Change after Addition of Boric Acid to Aqueous HCA Solution

(a) By fitting the experimental data in Figure 1, Appendix A for the change in the pH with the addition of boric acid for HCA concentrations of 0.1 mol/kg were obtained. 

The derivatives of Appendix A provide Appendix A for the rates of change of the pH with the addition of boric acid for HCA concentrations of 0.1 mol/kg.

As can be seen from Figure 1 and Appendix A, with the increase in the boric acid addition, the pH of the HCA aqueous solution decreased, that is [H^+^] in the solution increased. From Appendix A, it can be seen that the pH rates of change in the HCA solutions were negative, and with the increase in the boric acid dosage, the absolute value gradually decreased and then increased. Figure 4 shows the pH rate of change curves of the HCA with the change in the boric acid dosage. The absolute values of the pH rates of change in the HCA solutions had minimum values, which occurred when the molar ratio of boric acid to HCA was about 2. The actual ratios for glycolic acid, lactic acid, mandelic acid, gluconic acid, malic acid, L-tartaric acid, d-tartaric acid, and citric acid were 2.3, 2.0, 2.1, 1.9, 2.5, 2.1, 2.1, and 2.0, respectively.

(b) By fitting the experimental data in Figure 2, Appendix A for the change in the pH with the addition of boric acid for HCA concentrations of 0.2 mol/kg were obtained. 

The derivatives of Appendix A provide Appendix A for the rates of change of the pH with the addition of boric acid for HCA concentrations of 0.2 mol/kg.

As can be seen from Figure 2 and Appendix A, with the increase in the boric acid addition, the pH values of the HCA aqueous solutions decreased, that is, [H^+^] in the solution increased. From Appendix A, it can be seen that the absolute values of the pH rates of change in the HCA solutions gradually decreased and then increased. Figure 5 shows the pH rate of change curves of the HCA solutions with the amount of boric acid. The absolute values of the pH rates of change in the HCA solutions had minimum values, and the molar ratios of boric acid to HCA corresponding to the minimum values for the glycolic acid, lactic acid, mandelic acid, gluconic acid, malic acid, and D-tartaric acid were 3.5, 2.0, 2.0, 2.0, 2.1, and 2.2, respectively. The minimum rates of change in the pH for L-tartaric acid and citric acid were 0. When the rate of change in the pH for L-tartaric acid was 0, the molar ratios of boric acid to tartaric acid were 1.75 and 2.23, respectively. When the rate of change in the pH for citric acid was 0, the molar ratios of boric acid to citric acid were 1.46 and 2.37, respectively.

(c) By fitting the experimental data in Figure 3, Appendix A for the change in the pH with the addition of boric acid for HCA concentrations of 0.5 mol/kg were obtained. 

The derivatives of Appendix A provide Appendix A for the rates of change of the pH with the addition of boric acid for HCA concentrations of 0.5 mol/kg.

As can be seen from Figure 3 and Appendix A, with the increase in the boric acid addition, the pH of the HCA aqueous solution decreased, that is, [H^+^] in the solution increased. From Appendix A, it can be seen that the absolute values of the rates of change in the pH values of the HCAs decreased and then increased. Figure 6 shows the rates of change in the pH values of the HCAs with the amount of boric acid. The absolute values of the pH rates of change in the HCAs had minimum values, and the molar ratios of boric acid to HCA corresponding to the minimum values for the glycolic acid, lactic acid, mandelic acid, gluconic acid, malic acid, L-tartaric acid, and D-tartaric acid were 1.34, 0.75, 0.71, 0.68, 0.71, 0.69, and 0.68, respectively. The minimum pH rate of change for citric acid was 0, and the corresponding molar ratios of boric acid to citric acid were 0.62 and 0.94, respectively.

### 2.2. Comparison of Carboxylic Acids with Different Substituents

#### 2.2.1. Comparative Experiments with Different Substituent Carboxylic Acids

For comparison, carboxylic acids with different structures such as glycine, benzoic acid, salicylic acid, and oxalic acid were also investigated (Figure 7). The results showed that the pH of the boric acid–benzoic acid mixture did not change with an increase in the concentration of boric acid, while the pH of the reaction mixture for the other three carboxylic acids decreased slightly. Under the experimental conditions, the total pH rates of change in the reaction mixture after the addition of boric acid decreased in the order of glycine (9.7%) > salicylic acid (5.7%) > oxalic acid (2.3%) > benzoic acid (0). Because the pKa of boric acid was similar to that of glycine, the pH change in the boric acid–glycine mixture with the addition of boric acid did not necessarily suggest the formation of a strong complex between glycine and boric acid. Benzoic acid did not contain a hydroxyl group that could form an annular complex with boric acid, which explained the absence of an observable effect of boric acid addition on the pH of the boric acid–benzoic acid mixture. The hydroxyl and carboxyl groups of salicylic acid were at the ortho positions to each other on the benzene ring. Due to the influence of the benzene ring, the boric acid–salicylic acid mixture exhibited a greater rate of pH change compared to boric acid–oxalic acid.

#### 2.2.2. Influence of HCA Molecular Structure

The effect of boric acid on the pH of the reaction mixture was related to the number of groups in the HCA molecule that could bind to the boron atom and the stability of the bonding. For 0.1 and 0.2 mol/kg of HCAs, the effect of boric acid on the pH of the reaction mixture decreased in the following order: citric acid (the total pH rates of change was 24.5% and 39.6%) > L(+)-tartaric acid = D(−)-tartaric acid (21.9% and 33.3%) > D-gluconic acid (21.2% and 33.7%) > (R)-(−)-mandelic acid (19.8% and 32.4%) > L-(−)-malic acid (15.8% and 28.4%) > D-(−)-lactic acid (9.7% and 18.0%) > glycolic acid (5.4% and 11.8%), as shown in Figure 1 and Figure 2. For the HCAs with a concentration of 0.5 mol/kg, as the molar ratio of boric acid to HCAs increased, boric acid approached saturation in the water, and its effect on the pH decreased slightly following the order of citric acid (44.7%) > L(+)-tartaric acid = D(−)-tartaric acid (43.9%) > D-gluconic acid (44.1% and 33.7%) > (R)-(−)-mandelic acid (40.0%), with very similar the total pH rates of change (Figure 3). As indicated in Figure 1, Figure 2 and Figure 3, the effect of boric acid on the pH of boric acid–HCA mixture increased with increasing HCA concentration, where the total pH rates of change showed a similar trend among the seven HCAs, except for gluconic acid.

As shown by the above ranking of the total pH rates of change, the more hydroxyl groups in the HCA molecule, the greater the total pH rates of change. Specifically, the total pH rates of change of polycarboxylic acids such as citric acid and tartaric acid was greater than those of monobasic acids such as lactic acid and glycolic acid. The total pH rates of change in gluconic acid with multiple hydroxyl groups was greater than those of monobasic acids such as mandelic acid, lactic acid, and glycolic acid. Gluconic acid has five hydroxyl groups on its alkyl chain, which were prone to hydrogen bonding and could react with multiple boronic acids to form complexes. Consequently, the total pH rates of change increased significantly with an increasing concentration of gluconic acid, with the total pH rates of change at 0.5 mol/kg being similar to that of citric acid.

However, due to the strong electronegativity of the boron atom, HCAs with groups exhibiting strong electron-donating abilities could form stable complexes; thus, they would more readily undergo proton dissociation in aqueous solutions. Mandelic acid contains one benzene ring linked to the α-carbon atom, leading to a high electron cloud density on the benzene ring, causing the complexes formed by mandelic acid with boric acid to be highly stable. With the exception of gluconic acid, the reaction mixture containing mandelic acid showed the highest rate of pH change among the monobasic acids, which even surpassed that of malic acid, a dibasic carboxylic acid. By contrast, the reaction mixture containing lactic acid had a greater rate of pH change than glycolic acid because lactic acid contained a methyl group, which was a stronger electron-donating group.

Two chiral enantiomers, L-(+)-tartaric acid and D-(−)-tartaric acid, were selected for comparison. Boric acid was found to have the same deprotonation capacity for each tartaric acid enantiomer. Except for gluconic acid, the order of acidity of the boric acid and HCA mixtures was the same as the order of the total pH rate of change. The order from strong to weak was citric acid > tartaric acid > mandelic acid > malic acid > gluconic acid > lactic acid > glycolic acid.

### 2.3. Complexation Reaction of Boric Acid with HCA

The addition of boric acid could significantly improve the ionization ability of an HCA in an aqueous solution. It can be seen from Figure 4, Figure 5 and Figure 6 that with the increase in the molar ratio of boric acid to HCA, the absolute values of the pH rates of change in the HCAs gradually decreased and then slightly increased. The minimum values corresponded to molar ratios of boric acid to HCA of about 2 at HCA concentrations of 0.1 and 0.2 mol/kg and about 0.7 at HCA concentrations of 0.5 mol/kg. There should be some equilibrium at the absolute minimum of the pH rate of change. Boric acid and HCA can form 1:1 and 1:2 complexes, and large excesses of HCA are often required to facilitate the formation of 1:2 complexes. However, the results for both 1:1 and 1:2 complexes promote the ionization of HCA^−^ in aqueous solutions. If H_n_CA (n = 1, 2, 3) represents one, two, and three carboxylic alpha-hydroxyl carboxylic acids, respectively, (HO)_2_BCA^−^ and $B{\left(CA\right)}_{2}^{-}$ represent the 1:1 and 1:2 complexes formed by boric acid and HCA, respectively.

I: For HCAs containing carboxyl groups, the ionization process of the HCA aqueous solution after adding boric acid is as follows:(1)$$HCA\overset{k_{0}}{↔}CA^{-}+H^{+},$$
(2)$$H_{3}BO_{3}+CA^{-}\overset{k_{1}}{↔}(HO{)}_{2}BCA^{-}+H_{2}O,$$
(3)$${\left(HO\right)}_{2}BCA^{-}+HCA\overset{k_{2}}{↔}B{\left(CA\right)}_{2}{}^{-}+2H_{2}O,$$
where *k*_0_ is the ionization constant of HCA, *k*_1_ generates the equilibrium constant of the 1:1 monoligand complex (HO)_2_BCA^−^, and *k*_2_ generates the equilibrium constant of the 1:2 biligand complex $B{\left(CA\right)}_{2}^{-}$.

Assuming half of the HCA is fully ionized, the pH values corresponding to 0.1, 0.2, and 0.5 mol/kg concentrations were calculated to be about 1.3, 1.0, and 0.6, respectively. As can be seen from Figure 1, Figure 2 and Figure 3, the pH values of the HCA after the addition of boric acid were greater than the pH values calculated using the above assumptions. Therefore, the solution contained HCA, CA^−^, H_3_BO_3_, (HO)_2_BCA^−^, and $B{\left(CA\right)}_{2}^{-}$. Even if the molar ratio of boric acid to HCA reached 3, the solution still contained more un-ionized HCA.

(1) If we assume *k*_2_ << *k*_1_, then $\frac{k_{2}}{k_{1}}=\frac{\left[B{\left(CA\right)}_{2}{}^{-}]\;\times \;[CA^{-}]\;\times \;[H_{3}BO_{3}\right]}{[HCA]\;\times \;{\left[{\left(HO\right)}_{2}BCA^{-}\right]}^{2}}\to 0$ indicates that $[B(CA{)}_{2}{}^{-}]\to 0$, and less of the resulting monoligand is reconverted to diligand. That is, [(HO)_2_BCA^−^] >> [$B{\left(CA\right)}_{2}^{-}$] if [$B{\left(CA\right)}_{2}^{-}$] is ignored. Assuming that *k*_0_ << *k*_1_, the total reaction equation can be obtained from Formulas (2) and (3) as follows:(4)$$HCA+H_{3}BO_{3}\overset{K2}{↔}{\left(HO\right)}_{2}B{\left(CA\right)}^{-}+H^{+}+H_{2}O.$$

According to Formula (4), the equation for calculating the equilibrium constant *K*_2_ when the complexation product of boric acid and HCA is a single ligand is as follows:(5)$$K_{2}=\frac{\left[{\left(HO\right)}_{2}BCA^{-}]\times [H^{+}\right]}{\left[HCA]\times [H_{3}BO_{3}\right]}=k_{0}\times k_{1}.$$

By taking the logarithm of Formula (5), the following can be obtained:(6)$$\log K_{2}=\log[H^{+}]+\log[{\left(HO\right)}_{2}BCA^{-}]-\log[HCA]-\log[H_{3}BO_{3}],$$
where *K*_2_ is the total ionization equilibrium constant, *k*_0_ is the ionization equilibrium constant of the HCA, and *k*_1_ is the equilibrium constant of the complex reaction of boric acid and CA^−^ to form a single ligand.

From the data given in Figure 1, Figure 2 and Figure 3, we know the specified concentration of HCA (*C*_0_), the pH without the addition of boric acid (*y*_0_), and the quantity with the addition of boric acid (*x*_1_). As long as the value of the pH (*y*_1_) after adding boric acid is measured, $[H^{+}]=10^{-y_{1}}$, $[HCA]=C_{0}-10^{-y_{1}}$, $[{\left(HO\right)}_{2}BCA^{-}]=10^{-y_{1}}$, and $[H_{3}BO_{3}]=x_{1}-10^{-y_{1}}$ can be obtained using the formula $pH=-\log[H^{+}]$. Then, the total ionization equilibrium constant *K*_2_ and log*K*_2_ can be calculated. According to the formula $pKa=-\log K_{2}$, the acidity coefficient of the HCA after adding boric acid can be obtained:(7)$$pKa=-\log K_{2}=2y_{1}+\log(C_{0}-10^{-y_{1}})+\log(x_{1}-10^{-y_{1}}).$$

By measuring the pH (*y*_1_) value corresponding to a certain amount of boric acid (*x*_1_), Equation (7) can be used to obtain the acidity coefficient *pKa* when the complex product is a single ligand. Formula (7) is applicable for the case where the molar ratio of boric acid to HCA is large when the concentration of [H^+^] is large. Ragnar et al. studied the infrared spectra of complexes in an aqueous solution of lactic acid and boric acid, and they found that the molar ratios of boric acid and lactic acid were 0.4, 1.5, and 2. Through analysis, they found that boric acid and lactic acid formed a 1:1 complex when pH was 2 [4].

(2) If we assume that *k*_1_ << *k*_2_, then $\frac{k_{1}}{k_{2}}=\frac{[HCA]\;\times \;{\left[{\left(HO\right)}_{2}BCA^{-}\right]}^{2}}{\left[CA^{-}]\;\times \;[H_{3}BO_{3}]\;\times \;[B{\left(CA\right)}_{2}{}^{-}\right]}\to 0$ indicates that $[{\left(HO\right)}_{2}BCA^{-}]\to 0$, that is, the generated single ligand is quickly converted to a double ligand. Then, Formulas (2) and (3) can be combined into the following:(8)$$H_{3}BO_{3}+HCA+CA^{-}\overset{{k^{\prime }}_{1}}{↔}B{\left(CA\right)}_{2}{}^{-}+H_{2}O.$$

Assuming that *k*_0_ << *k*′_1_, $[CA^{-}]\to 0$, the total reaction equation can be obtained from (1) and (8) as follows:(9)$$2HCA+H_{3}BO_{3}\overset{K1}{↔}B{\left(CA\right)}_{2}{}^{-}+H^{+}+2H_{2}O.$$

According to Formula (9), the equation for calculating the equilibrium constant *K*_1_ when the complexation product of boric acid and HCA is a double ligand is as follows:(10)$$K_{1}=\frac{\left[B{\left(CA\right)}_{2}{}^{-}]\times [H^{+}\right]}{{\left[HCA\right]}^{2}\times [H_{3}BO_{3}]}=k_{0}\times k{’}_{1},$$
where *K*_1_ is the total ionization equilibrium constant, *k*_0_ is the ionization equilibrium constant of HCA, and *k*′_1_ is the equilibrium constant of boric acid combining with CA^−^ to form a double ligand. By taking the logarithm of Formula (10), the following can be obtained:(11)$$\log K_{1}=\log[H^{+}]+\log[B{\left(CA\right)}_{2}{}^{2-}]-2\log[HCA]-\log[H_{3}BO_{3}].$$

From the data shown in Figure 1, Figure 2 and Figure 3, we know the specified concentration of HCA (*C*_0_), the pH (*y*_0_) without the addition of boric acid, and the quantity with the addition of boric acid (*x*_1_). As long as the pH (*y*_1_) after adding boric acid is measured, $[H^{+}]=10^{-y_{1}}$, $[HCA]=C_{0}-2\times 10^{-y_{1}}$, $[B{\left(CA\right)}_{2}{}^{-}]=10^{-y_{1}}$, and $[H_{3}BO_{3}]=x_{1}-10^{-y_{1}}$ can be obtained using the formula $pH=-\log[H^{+}]$, and then the total ionization equilibrium constant *K*_1_ and log*K*_1_ can be calculated. According to the formula $pKa=-\log K_{1}$, the acidity coefficient of the HCA after adding boric acid can be obtained as follows:(12)$$pKa=-\log K_{1}=2y_{1}+2\log(C_{0}-2\times 10^{-y_{1}})+\log(x_{1}-10^{-y_{1}}).$$

By measuring the pH (*y*_1_) corresponding to a certain amount of boric acid (*x*_1_), Equation (12) can be used to obtain the acidity coefficient *pKa* when the complex product is a double ligand. Formula (12) is applicable when the concentration of [H^+^] is large, the ionization equilibrium constant of HCA is small, and the molar ratio of HCA to boric acid is large. Maseda et al. used a concentration of 1.0 mol/dm^3^ of lactic acid and a concentration of 0.02 mol/dm^3^ of boric acid to adjust the pH of the aqueous solution to ≤2.5 and determined the formed diligand through 11B NMR analysis [6].

For a certain concentration of HCA, with the increase in the molar ratio of boric acid to HCA, the concentration of boric acid increases, which promotes the equilibrium given by Equation (2) to move in the positive direction, that is, the content of a single-ligand complex increases. At the same time, the equilibrium of Equation (1) moves in the positive direction, that is, the relative content of the HCA decreases. Since the amount of increase in the single-ligand complex is the same as the amount of decrease in the HCA, the effect of this on the equilibrium of Equation (3) is bidirectional. Therefore, when the concentration of boric acid is low, two-ligand-dominated complexes are formed, whereas when the concentration of boric acid is increased, the relative content of the single-ligand complex increases faster. From Figure 4, Figure 5 and Figure 6, it can be seen that the molar ratio of boric acid to HCA corresponding to the minimum value of the pH rate of change was about 2 for HCA concentrations of 0.1 and 0.2 mol/kg. When the concentration of HCA was increased to 0.5 mol/kg, the molar ratio of boric acid to HCA decreased to about 0.7, corresponding to the minimum pH rate of change. With increasing HCA concentrations, the association between HCA molecules was enhanced, and the complexation reaction between boric acid and HCA was more sensitive to the effect of pH change rate. At the same time, as the boric acid concentration increased, the molar ratio of boric acid to HCA corresponding to the minimum pH change rate decreased correspondingly.

For gluconic acid containing more than one hydroxyl group, the pH rate of change was larger than those of malic acid and tartaric acid. The complex reaction with boric acid was more complex than those of glycolic acid and lactic acid. In addition to the carboxyl group and α hydroxyl group, the other four hydroxyl groups could also form complexes with boric acid. With the increase in the number of types of complexes formed by gluconic acid and boric acid, the ionization equilibrium of gluconic acid moves further in the forward direction, and the concentration of [H^+^] is much higher than that without boric acid.

II: For an H_2_CA, such as tartaric acid and malic acid, the ionization and possible complexation reactions of the H_2_CA aqueous solution after adding boric acid are as follows:(13)$$H_{2}CA\overset{k_{0}}{↔}HCA^{-}+H^{+},$$
(14)$$HCA^{-}\overset{k_{1}}{↔}CA^{2-}+H^{+},$$
(15)$$H_{3}BO_{3}+HCA^{-}\overset{k_{2}}{↔}{\left(HO\right)}_{2}BHCA^{-}+H_{2}O,$$
(16)$${\left(HO\right)}_{2}BHCA^{-}+H_{2}CA\overset{k_{3}}{↔}(H_{2}CA)BHCA^{-}+2H_{2}O,$$
(17)$${\left(HO\right)}_{2}BHCA^{-}+HCA^{-}\overset{k_{4}}{↔}B{\left(HCA\right)}_{2}{}^{2-}+2H_{2}O,$$
(18)$${\left(HO\right)}_{2}BHCA^{-}+CA^{2-}\overset{k_{5}}{↔}{\left(HCA\right)}^{-}B{\left(CA\right)}^{2-}+2H_{2}O,$$
(19)$$H_{3}BO_{3}+CA^{2-}\overset{k_{6}}{↔}{\left(HO\right)}_{2}BCA^{2-}+H_{2}O,$$
(20)$${\left(HO\right)}_{2}BCA^{2-}+H_{2}CA\overset{k_{7}}{↔}(H_{2}CA)BCA^{2-}+2H_{2}O,$$
(21)$${\left(HO\right)}_{2}BCA^{2-}+CA^{2-}\overset{k_{8}}{↔}B{\left(CA\right)}_{2}{}^{4-}+2H_{2}O,$$
where *k*_0_ and *k*_1_ are the ionization equilibrium constants for the aqueous H_2_CA, and *k*_2_–*k*_8_ are the complexation reaction equilibrium constants. Equations (18) and (21) are possible complexation reactions following the ionization of tartaric acid.

As can be seen from Equations (13)–(21), when boric acid was added to the H_2_CA aqueous solution, the number of complex species that could be formed was much higher than those of the HCAs containing one carboxyl group, which means that more anions ionized by H_2_CA could be consumed. Thus, the pH rate of change was greater than those of the HCAs containing one carboxyl group.

III: Citric acid contains three carboxyl groups. After adding boric acid, the ionization and possible complexation reactions of the citric acid solution are as follows:(22)$$H_{3}CA\overset{k_{0}}{↔}H_{2}CA^{-}+H^{+},$$
(23)$$H_{2}CA^{-}\overset{k_{1}}{↔}HCA^{2-}+H^{+},$$
(24)$$HCA^{2-}\overset{k_{2}}{↔}CA^{3-}+H^{+},$$
(25)$$H_{3}BO_{3}+H_{2}CA^{-}\overset{k_{3}}{↔}{\left(HO\right)}_{2}BH_{2}CA^{-}+H_{2}O,$$
(26)$${\left(HO\right)}_{2}BH_{2}CA^{-}+H_{3}CA\overset{k_{4}}{↔}(H_{3}CA)BH_{2}CA^{-}+2H_{2}O,$$
(27)$${\left(HO\right)}_{2}BH_{2}CA^{-}+H_{2}CA^{-}\overset{k_{5}}{↔}B{\left(H_{2}CA\right)}_{2}{}^{2-}+2H_{2}O,$$
(28)$${\left(HO\right)}_{2}BH_{2}CA^{-}+HCA^{2-}\overset{k_{6}}{↔}{\left(H_{2}CA\right)}^{-}B{\left(HCA\right)}^{2-}+2H_{2}O,$$
(29)$$H_{3}BO_{3}+HCA^{2-}\overset{k_{7}}{↔}{\left(HO\right)}_{2}BHCA^{2-}+H_{2}O,$$
(30)$${\left(HO\right)}_{2}BHCA^{2-}+HCA^{2-}\overset{k_{8}}{↔}B{\left(HCA\right)}_{2}{}^{4-}+2H_{2}O,$$
(31)$${\left(HO\right)}_{2}BHCA^{2-}+H_{3}CA\overset{k_{9}}{↔}(H_{3}CA)B{\left(HCA\right)}^{2-}+2H_{2}O,$$
(32)$$H_{3}BO_{3}+CA^{3-}\overset{k_{10}}{↔}{\left(HO\right)}_{2}BCA^{3-}+H_{2}O,$$
(33)$${\left(HO\right)}_{2}BCA^{3-}+H_{2}CA^{-}\overset{k_{11}}{↔}{\left(H_{2}CA\right)}^{-}B{\left(CA\right)}^{3-}+2H_{2}O,$$
(34)$${\left(HO\right)}_{2}BCA^{3-}+HCA^{2-}\overset{k_{12}}{↔}{\left(HCA\right)}^{2-}B{\left(CA\right)}^{3-}+2H_{2}O,$$
(35)$${\left(HO\right)}_{2}BCA^{3-}+H_{3}CA\overset{k_{13}}{↔}(H_{3}CA)BCA^{3-}+2H_{2}O,$$
where *k*_0_–*k*_3_ are the ionization equilibrium constants for aqueous citric acid solutions, and *k*_4_–*k*_13_ are the complexation reaction equilibrium constants.

As can be seen from Equations (22)–(35), complex complexation reactions can occur after the addition of boric acid in a citric acid solution, and more-complex species can be formed compared with those for HCAs containing two carboxyl groups, which means that more H_3_CA can be consumed. Thus, the pH rate of change was greater than those of the HCAs containing two carboxyl groups. Since the ionization constants of citric acid are pK_1_ = 3.13, pK_2_ = 4.76, and pK_3_ = 6.40, the content of CA^3−^ in an aqueous solution of citric acid would be very low. The concentration of the complexation product [BCA^3−^] of Equation (32) was estimated to be very low under an acidic environment, and the concentrations of complexation products of Equations (33)–(35) were likely to be even lower.

When an HCA containing multiple carboxyl groups, such as tartaric acid, malic acid, and citric acid, is ionized in an aqueous solution, if only first-order ionization is taken into account and it is assumed that only single- or double-ligand complex products with negative charges are formed with boric acid, then its ionization equilibrium constant can be obtained from Formulas (5) and (10). The acidity coefficient of HCA after adding boric acid was calculated by Formulas (7) and (12), and the corresponding acidity coefficients at the apexes of the curves in Figure 4, Figure 5 and Figure 6 are shown in Table 1. When boric acid was added to the HCA aqueous solutions, the acidity coefficients of the solutions with monoligand and diligand complexes are shown as pKa values.

From Formulas (7) and (12), it can be seen that the acidity coefficient of HCA after adding boric acid is a function of three variables: the HCA concentration (C_0_), the molar ratio of boric acid to HCA (x), and the measured value of the solution pH (y). As can be seen from Table 1, when C_0_, x, and y were the same, the pKa (1:1) calculated by the single-ligand formula was greater than that calculated by the double-ligand formula (1:2). There were various complexation reactions in the HCA aqueous solution after the addition of boric acid. Usually, monoligands and diligands coexist, and the actual pKa value should be between the two, that is, pKa (1:2) < pKa < pKa (1:1).

According to Formulas (1)–(35), as the molar ratio of boric acid to HCA (x) increased, the composition of the complex in the aqueous solution also changed. Combined with the analysis in Table 1, it can be seen that when x was very small, boric acid and HCA formed a 1:2 double-ligand complex, and the corresponding pH rate of change was large. When x increased, the content of the 1:1 monoligand complex formed by boric acid and HCA increased, the pH of the solution decreased, as did the pH rate of change. When x was increased to the minimum pH rate of change, the rate of increase in the 1:2 complex amount in the aqueous solution was close to 0. As x continued to increase, the content of the monoligand complex continued to increase, which was represented by a slight increase in the pH rate of change. However, as boric acid approached saturation, the pH did not change with x or showed only small fluctuations (measurement error).

### 2.4. Fatty Acid Esterification Catalyzed by Boric Acid–HCA Complexes

Concentrated sulfuric acid has often been used as a catalyst for esterification of fatty acids to prepare fatty acid esters. However, the use of concentrated sulfuric acid could result in the corrosion of equipment and formation of a large amount of wastewater. Moreover, its strong oxidizing ability may lead to darkening of the products. By contrast, the mixture of boric acid and HCAs could catalyze the esterification of fatty acids with high product selectivity and without product darkening.

The effect of the composition of the boric acid–HCA mixture as a catalyst system for the synthesis of methyl palmitate is shown in Figure 8. When boric acid was used alone as the catalyst, the conversion of palmitic acid after 8 h of reaction was only 7.2%; however, the conversion of palmitic acid increased to 85% after the addition of tartaric acid (Figure 8a). As indicated by Figure 8b, the 20 h conversion of palmitic acid was 44.5% when using tartaric acid alone as the catalyst, but the addition of boric acid increased the yield of methyl ester to 98%. Hence, the mixture of boric acid and tartaric acid had better catalytic performance than boric acid or tartaric acid alone, which was mainly due to the enhanced deprotonation of tartaric acid in the presence of boric acid.

Because the esterification reaction produced water, it was advantageous to use B_2_O_3_ instead of boric acid as the catalyst. Excess alcohol also favored the formation of esters and, in addition to affecting the reaction equilibrium, more importantly, promoted the dissolution of the catalyst in the system. A notable advantage of such a reaction system was that when the methyl esterification reaction was complete, the reaction mixture could be left to settle into two distinct layers, with the catalyst being predominantly present in the methanol layer and the main product (fatty acid methyl ester) being neutral in polarity. This was advantageous for both the recycling of the catalyst and the refining of the product, making such a reaction system suitable for large-scale industrial applications.

The mass spectrum of methyl palmitate is shown in the Appendix A. The infrared spectrum of methyl palmitate is shown in the Appendix A. It can be seen from Appendix A that 2923.60 cm^−1^ was the antisymmetric stretching vibration peak of C-H in -CH_2_-. 2852.25 cm^−1^ was the symmetric stretching vibration peak of -CH_2_-. 1743.36 cm^−1^ was the stretching vibration peak of ester > C=O. 1465.91 cm^−1^ and 1436.05 cm^−1^ were the superimposed peaks of -CH_3_ asymmetric deformation vibration and -CH_2_ shear vibration. 1363.04 cm^−1^ was the bending vibration absorption peak of CH_3_. 1196.3 cm^−1^, 1170.60 cm^−1^ was the antisymmetric stretching vibration peak of ester C-O-C. 

## 3. Experimental Section

### 3.1. Materials and Apparatus

The following starting materials and reagents were used in this study: boric acid (98%), glycolic acid (98%), citric acid (99.5%), L (+)-tartaric acid (99.5%), D-(−)-tartaric acid (99.5%), D-(−)-lactic acid (90%), (R)-(−)-mandelic acid (99%), L-(−)-malic acid, D-gluconic acid (49–53 wt.% in H_2_O), oxalic acid (99%), palmitic acid (97%), boron oxide (98%), and phenylboronic acid (97%). These compounds were purchased from Macklin and Aladdin (Shanghai, China). Methanol (99.5%), sodium hydroxide (98%), and sodium carbonate anhydrous (98%) were purchased from Chengdu Kelong Chemical (Chengdu, Sichuan Province, China). Distilled water was prepared in the laboratory.

The reaction apparatus was an organic synthesis unit PPV-3000 (EYELA, Tokyo Rikakikai, Tokyo, Japan). The following analytical instruments were used in this work: an LC-PH-3L desktop pH meter (Lichen, Shanghai, China) with a resolution of 0.01 and a 201-composite electrode, a near-infrared quality analyzer, SY-3650-II, Foss NIRSystems Inc. (Laurel, MD, USA), an AVANCE III 300 MHz or 600 MHz nuclear magnetic resonance spectrometer (Bruker, Fällanden, Switzerland); a 7890A gas chromatograph (Agilent, Santa Clara, CA, USA) equipped with quartz capillary chromatography columns (60 m × 0.25 mm × 0.25 μm) with AT-35 as the immobile phase; a TQ456 GC-MS instrument (Bruker, Billerica, MA, USA) equipped with BR-5 elastic quartz capillary columns (30 m × 0.25 mm × 0.25 μm) as chromatography columns.

### 3.2. Experimental Methods

#### 3.2.1. Determination of the pH of Boric Acid/HCA Mixtures

The HCAs were prepared at three different concentrations of 0.1, 0.2, and 0.5 mol/kg with distilled water. Then, boric acid was added at different proportions to three replicate HCA solutions of each given concentration, followed by measuring the pH values of the three replicate solutions. The total pH rates of change in the HCA solutions were calculated by
(pH_0_ − pH_1_)/pH_0_,
where pH_0_ is the initial pH of the HCAs at the given concentration, and pH_1_ is the pH after the addition of boric acid.

#### 3.2.2. Fatty Acid Ester Synthesis

In a reaction flask, 10 g of fatty acid, 20 g of methanol, 0–0.5 g of boric acid, and 0–2.5 g of tartaric acid were added and stirred magnetically (500 rpm). The reaction temperature was controlled at 65 °C, and the reaction time was 8–20 h. After the reaction, the product was poured into a separatory funnel and allowed to settle into two distinct layers. The layer of fatty acid ester was separated, washed with water, dried with anhydrous sodium sulfate, and then sampled for analysis.

### 3.3. Analytical Methods

For infrared spectral data acquisition, the sample was placed on an infrared spectrometer slide with air as the background, and the spectrum was acquired in the wave number range of 400–4000 cm^−1^, where the number of sample scans was 24, the number of background scans was 24, the sample gain was 1.0, the mirror velocity was 0.6329, and the aperture was 95.00.

For H-NMR data acquisition, the sample was placed into a measuring sample tube, CDCl_3_ was added, and the sample was measured by a 600 MHz NMR instrument (frequency: 600.18 MHz), where the temperature was 296.9 K, the number of scans was 64, and the pulse width was 12.6 μs. The spectral width was 12,315.27 and the data point size was 32,768. The NMR spectra were processed by Mestrenova software, and integrated after calibration, phase, and baseline calibration.

For GC analysis, high-purity nitrogen was used as the carrier gas, and the temperature program was as follows. The initial temperature was 70 °C (held for 2 min), with the first ramp of 5 °C/min to 150 °C (held for 3 min), followed by a second ramp of 10 °C/min to 230 °C (held for 10 min). The inlet temperature was set to 250 °C, and the total flow rate was set to 130.5 mL/min, with a split ratio of 50:1 and a septum purge rate of 3 mL/min. The analytes were detected using a flame ionization detector (FID), with a detection port temperature of 250 °C, a hydrogen flow rate of 40 mL/min, an air flow rate of 450 mL/min, and a nitrogen purge rate of 25 mL/min. The injection volume was 0.2 µL.

For GC-MS analysis, high-purity helium was used as the carrier gas, and the temperature program was as follows. The initial temperature was 50 °C (held for 3 min), with a first ramp of 20 °C/min to 120 °C, followed by a second ramp of 2 °C/min to 180 °C (held for 2 min), and a third ramp of 50 °C/min to 250 °C (held for 5 min). The inlet temperature was set to 230 °C, and the interface temperature was set to 250 °C.

For mass spectrometry, electron ionization (EI) was used as the ionization source, with an ionization voltage of 70 eV, where full-scan mode was used with a scan range of 45–350 amu. In addition, a solvent delay time of 5 min was set, where the injection volume was set to 0.5 μL (the sample was dissolved in ethanol with a mass fraction of 1%).

## 4. Conclusions

(1)The effect of boric acid on the ionization balance of HCA was studied by analyzing and measuring the pH values of aqueous solutions of eight HCAs—glycolic acid, D-(−)-lactic acid, (R)-(−)-mandelic acid, D-gluconic acid, L-(−)-malic acid, L-(+) -tartaric acid, D-(−)-tartaric acid, and citric acid—after adding boric acid. The functions for the pH variations of the HCAs with the amount of boric acid at specified concentrations were obtained by polynomial fitting. By differentiating the fitted pH functions, the functions of the pH rate of change with the change in the boric acid dosage at specified concentrations were obtained. The complexation reactions of boric acid and HCAs were analyzed, and the formulas for calculating the ionization equilibrium constants of monoligand complexes and diligand complexes were deduced. The effects of these two complexes on the ionization equilibrium were compared.(2)Boric acid can react with the hydroxyl groups of HCA to form complexes, which promotes the ionization equilibrium of HCA to move in the positive direction. HCA molecules contain strong electron donor groups, and the boric acid complex became more stable with a greater proton donating ability. The acidities of the combination of boric acid and HCAs were in the following order: citric acid > tartaric acid > mandelic acid > malic acid > grape acid > lactic acid > glycolic acid.(3)The compound catalyst composed of tartaric acid and boric acid was used to catalyze the esterification of palmitic acid and methanol, and the yield of methyl ester was up to 98%. The short-chain alcohol was favorable to promote the dissolution of the complex catalyst. The catalyst was mainly present in the alcohol after the esterification reaction was finished, and the product ester was neutral. This facilitated the recovery of catalyst and excess alcohol, which was beneficial to environmentally friendly production.

## Figures and Tables

**Figure 1 molecules-28-04723-f001:**
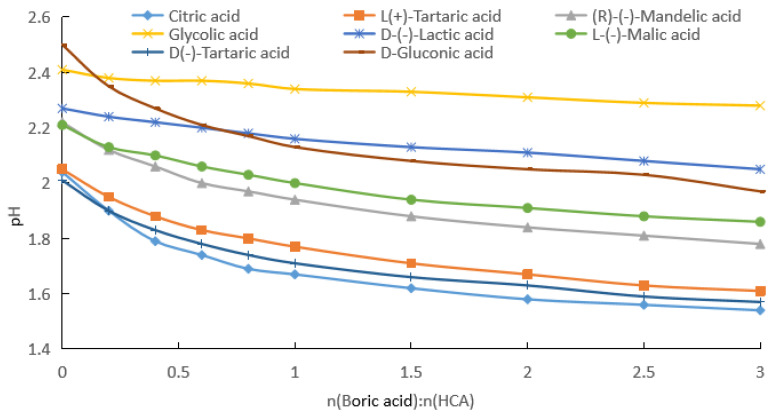
pH values of boric acid and hydroxy carboxylic acid (HCA) aqueous solutions, with HCA solubility of 0.1 mol/kg.

**Figure 2 molecules-28-04723-f002:**
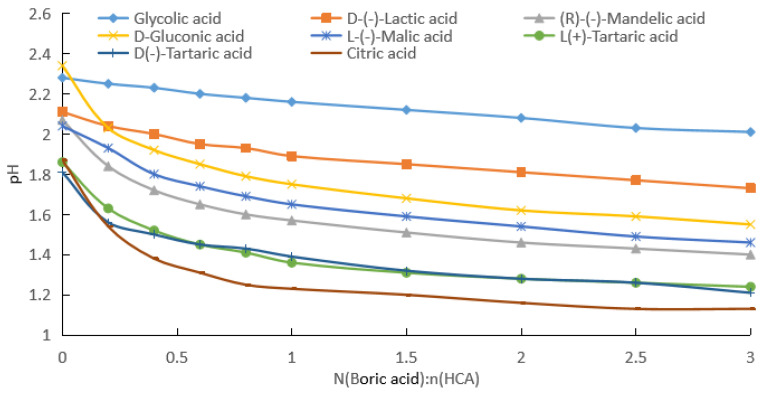
pH values of boric acid and HCA aqueous solutions, with HCA solubility of 0.2 mol/kg.

**Figure 3 molecules-28-04723-f003:**
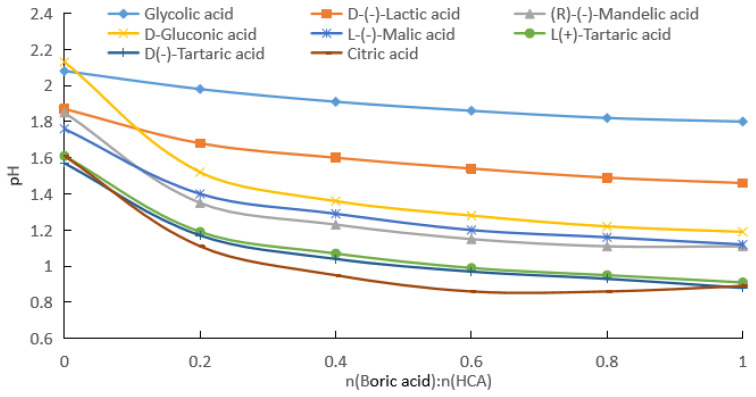
pH values of boric acid and HCA aqueous solution, with HCA solubility of 0.5 mol/kg.

**Figure 4 molecules-28-04723-f004:**
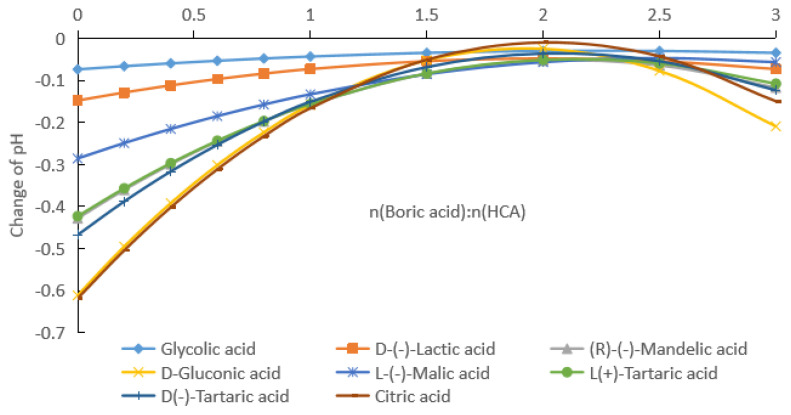
Rates of change pH with boric acid dosage at HCA concentrations of 0.1 mol/kg.

**Figure 5 molecules-28-04723-f005:**
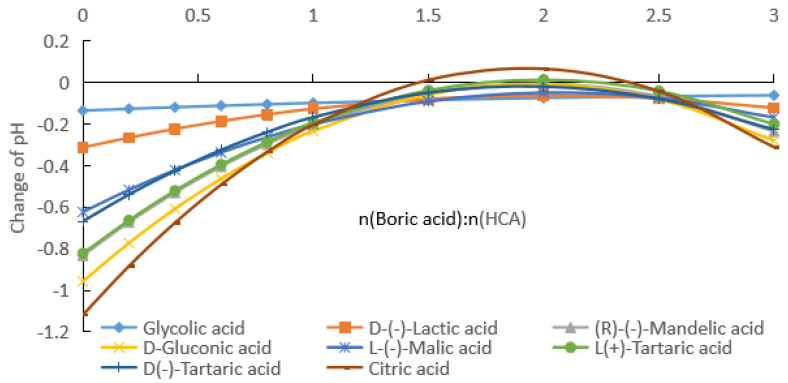
Rates of change in pH with boric acid dosage at HCA concentrations of 0.2 mol/kg.

**Figure 6 molecules-28-04723-f006:**
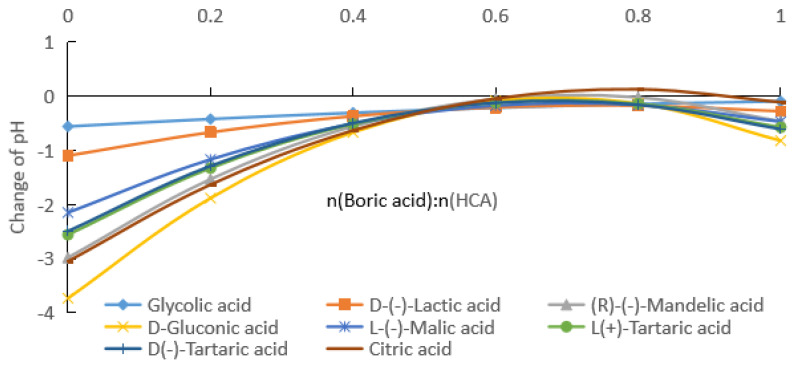
pH rates of change with boric acid dosage at HCA concentrations of 0.5 mol/kg.

**Figure 7 molecules-28-04723-f007:**
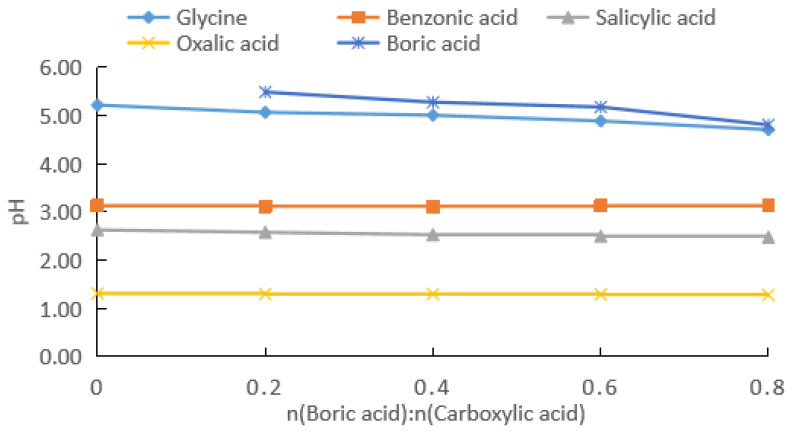
pH changes after adding boric acid and carboxylic acid, with carboxylic acid concentration of 0.1 mol/kg.

**Figure 8 molecules-28-04723-f008:**
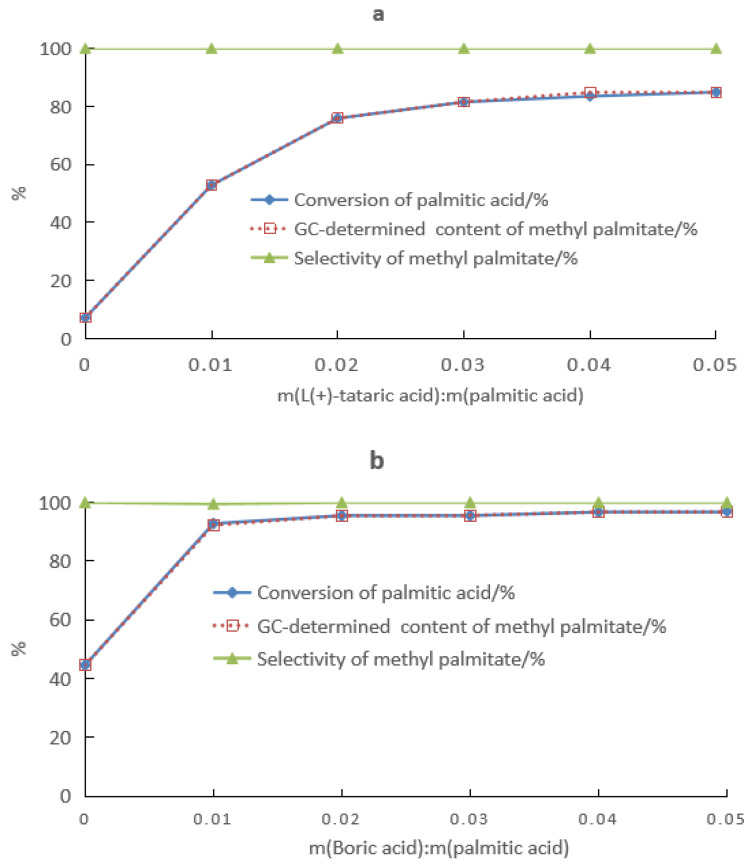
Effects of the dosage of (**a**) tartaric acid and (**b**) boric acid on the conversion of palmitic acid, and the gas chromatography (GC)-determined content and selectivity of the product methyl palmitate: (**a**) reaction conditions of m (palmitic acid):m (methanol):m (boric acid) = 10:20:0.5, 65 °C, and 8 h. (**b**) Reaction conditions of m (palmitic acid):m (methanol):m (tartaric acid) = 10:20:0.5, 65 °C, and 20 h.

**Table 1 molecules-28-04723-t001:** Acidity coefficients of boric acid and HCAs to form single or double ligands.

Potency (mol/kg)	pKa	GA	D-(−)LA	R-(−)MA	D-GlcA	L-(−)H_2_Mi	L-(+)TA	D-(−)TA	CA
0.1	pKa (1:1)	2.93	2.46	1.9	2.33	2.08	1.5	1.39	1.29
	pKa (1:2)	1.86	1.6	0.67	1.21	0.87	0.11	−0.05	−0.21
0.2	pKa (1:1)	3.53	2.47	1.72	2.09	1.89	1.28	1.84	1.09
	pKa (1:2)	2.3	1.66	0.74	1.22	0.96	0.08	0.89	−0.26
0.5	pKa (1:1)	3.05	2.22	1.36	1.61	1.47	0.93	0.87	0.72
	pKa (1:2)	2.7	1.83	0.83	1.14	0.97	0.26	0.17	−0.11

Note: Values computed where the pH rates of change were the smallest; pKa (1:1) is the acidity system assuming that only single ligands were formed; pKa (1:2) is the acidity coefficient assuming that only double ligands were formed.

## Data Availability

The data presented in this study are available on request from the corresponding author.

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
