# Peer review of "Effect of Boric Acid on the Ionization Equilibrium of α-Hydroxy Carboxylic Acids and the Study of Its Applications"

_molecules, 2023, doi:10.3390/molecules28124723_

Round 1

Reviewer 1 Report

The authors have given results as to the pH dependance of the molar ratio of boric acid in relation to various HCA's.  With this the authors decided to precent the results by first discussing the individual HCA's followed  by the kinetic evaluation and lastly the application to one methylation reaction with one of the HCA's.  

The first part of the publication as to the fitting (equations) of the data (figures) to the measured results the equations can be placed in 'n support document  for this I found not contributing further to the article. These are equations 1 - 40.  The graphs indicates the the needed information.  

In the abstract line 19 the word dosage is used, of which this may not be the best word to be used, this might work better ....with an increase in the boric acid molar ratio.... instead of dosage which is generally found in pharmacuetical environments.

Line 32 Earth can be earth it need not be capital letters.

line 58 inactivation should be deactivation.  A catalyst is deactivated or poisened rendering it inactive for further catalytic reactions.

line 74 pKa1 is it pKa1?

figures 1 -3 the legend of figure 1 is not the same as for 2 and 3.   This makes comparison difficult. Was the pH of citric acid done at 0.1 mol/kg?

As per line 200 the authors started to use the ratio of change where previously it was rate of change why the different choise of words?

Figure 7 the title of the figure is pH changes after adding boric acid and carboxylic acid, with a carboxylic acid concentration of.....  Does this imply that one has added aditional carboxcylic acid of only boric acid.  Then also in the figure there are only boric acid, which indicate that there is an drift in the pH reading which might be an indication of a measurement error, which is not indicated in any of the figures given.  From the experimental it stipulates that the pH meter can only measure to a 0.01 value.  If the this is the case and the changes in pH is smaller than what can be measured is the measured values accurate?  Please give an indication of the error value on the measurements done.

The statement in line 204 is incorrect, for boric acid indeed do form a complex with benzoic acid although it may not change the pH of the solution.  For there are reactions that shows this in literature.  It can be that due water as solvent this might change the environment somewhat.  Please revisit and revize if needed.

Line 288 lgK2  should be logK2

line 383 the sentence more-complex complex ..... does not read correct please check to confirm.

Line 445 the title for figure 8's reaction conditions: for a) four  reagents are given while for b) only three.  These are presumed to be the mass and not the mole or the mole ratios?

line 460 the units for the frequency for the 300 MHz and 600 MHz and not just M.

line 502 the u in amu is in red why?

Why was tartaric acid chosen for the methylation reaction even though the results showed citric acid to be the better substrate to use? 

Was the esterfication reaction investigated for other acids expept for palmitic acid?

To get more information as to the effect of pH and 11B NMR study could be be done which can support the results presented. A proton kinetic study in water can also give direct insight into the methyl ester formation.

No NMR spectra or GC (MS) data are presented.  It is presumed the chromatograms and spectra obtained were for the confimation of the methyl ester product?  Is this available for supporting information?

Was the IR in transmission mode using KBr or with an ATR?

Reviewer 2 Report

Meng et al reported their studies of effects of the complexation reactions between boric acid and HCAsa ionization equilibrium of the HCAs by analyzing and measuring the pH values of aqueous solutions of different common acids. Then The compound catalyst composed of tartaric acid and boric acid was used to catalyze the esterification of palmitic acid and methanol, and the yield of methyl ester was up to 98%, avoiding the use of corrosive and toxic  H2SO4, HCl, H3PO4, or organic sulfonic acids. This paper is reasonable and well studied. It can be accepted.

Minor things,

1.     The author didn’t provide the resource of boric acid, but they provided the source of B2O3 and phenylboronic acid, so what exactly B source.

2.     Activation of acid by boric acid has successfully applied in the synthetic organic chemistry. The related ref, such as J. Am. Chem. Soc. 2018, 140, 5899-5903 should be cited in the introduction.

Round 2

Reviewer 1 Report

Dear Authors thank you for your publication. 

Line 85 Just to clarify the figures the authors may include either in the heading or as a sentence

"The first order derivatives of .........."